# Allergic Anaphylactic Risk in Farming Activities: A Systematic Review

**DOI:** 10.3390/ijerph17144921

**Published:** 2020-07-08

**Authors:** Giulio Arcangeli, Veronica Traversini, Emanuela Tomasini, Antonio Baldassarre, Luigi Isaia Lecca, Raymond P. Galea, Nicola Mucci

**Affiliations:** 1Department of Experimental and Clinical Medicine, University of Florence, 50139 Florence, Italy; giulio.arcangeli@unifi.it (G.A.); nicola.mucci@unifi.it (N.M.); 2Occupational Medicine School, University of Florence, 50139 Florence, Italy; 3Department Medical and Surgical Specialties, Radiological Sciences, and Public Health, University of Brescia, 25121 Brescia, Italy; emanuela.tomasini@unibs.it; 4Doctoral School in Clinical Sciences, University of Florence, 50139 Florence, Italy; antonio.baldassarre@unifi.it (A.B.); luigiisaia.lecca@unifi.it (L.I.L.); 5Faculty of Medicine & Surgery, University of Malta, MSD 2080 Msida, Malta; raymond.galea@um.edu.mt; 6Head of the Malta Postgraduate Medical Training Programme, Mater Dei Hospital, MSD 2090 Msida, Malta

**Keywords:** allergy, anaphylaxis, anaphylactic shock, farm workers, agriculture, occupational medicine, exposure, risk, prevention

## Abstract

Allergic disorders in the agriculture sector are very common among farm workers, causing many injuries and occupational diseases every year. Agricultural employees are exposed to multiple conditions and various allergenic substances, which could be related to onset of anaphylactic reactions. This systematic review highlights the main clinical manifestation, the allergens that are mostly involved and the main activities that are usually involved. This research includes articles published on the major databases (PubMed, Cochrane Library, Scopus), using a combination of keywords. The online search yielded 489 references; after selection, by the authors, 36 articles (nine reviews and 27 original articles) were analyzed. From this analysis, the main clinical problems that were diagnosed in this category were respiratory (ranging from rhinitis to asthma) and dermatological (eczema, dermatitis, hives) in nature, with a wide symptomatology (from a simple local reaction to anaphylaxis). The main activities associated with these allergic conditions are harvesting or cultivation of fruit and cereals, beekeepers and people working in greenhouses. Finally, in addition to the allergens already known, new ones have emerged, including triticale, wine, spider and biological dust. For these reasons, in the agricultural sector, research needs to be amplified, considering new sectors, new technologies and new products, and ensuring a system of prevention to reduce this risk.

## 1. Introduction

Anaphylaxis is the most serious form of an allergic reaction, defined as “a serious, life-threatening generalized or systemic hypersensitivity reaction” that is “rapid in onset and might cause death” [1]. It may occur following exposure to various triggering factors, or be due to no apparent triggers (idiopathic anaphylaxis). Signs and symptoms of anaphylactic reaction involve the respiratory, cardiovascular, gastrointestinal systems, skin and mucosal changes, such as dyspnea, tachycardia, nausea, abdominal pain and urticarial rashes [1,2,3]. 

The global burden of this phenomenon is still uncertain [4], because this syndrome is often “under-recognized” or “under-diagnosed” [1]. During the last few years, various international studies have shown an upward trend in the rate of anaphylaxis, even if the incidence and prevalence differ between the European and American populations, especially regarding the occurrence of fatal outcomes. In Europe, the calculated prevalence is currently around 3%, while in the United States it ranges between 0.05–2% [4]. The increase in the frequency recorded globally appears to be conditioned not only by changes in the diagnostic definition and management of the event, but also by many different variables, such as triggers, socio-demographic dates, geographical area, seasonal variations, older age, gender, atopy and comorbidities (e.g., allergic asthma, cardiovascular disease) [4]. In the work environment, some of the main provocative agents, (together with others, which are less common) can cause an allergic response, exposing workers to a significant risk factor, which is likely to arouse a severe or fatal anaphylactic reaction. In work-related anaphylaxis, causes and conditions may be directly related to the workplace [5]; otherwise, a contemporary or preexisting allergy (e.g., food/pollen allergy) can be triggered by occupational allergens (work-exacerbated anaphylaxis) [6]. Occupational anaphylaxis may develop from the same agents responsible for occupational asthma or generalized urticarial reactions [5]. These allergens may enter the body via inhalation, skin-contact, bites or stings and accidental ingestion of contaminants in work environments that do not comply with the necessary industrial hygiene measures. Cofactors—such as physical exercise, fever, acute infection, emotional stress, etc.—increase the risk of induction or exacerbation of an anaphylactic reaction [2]. The most common cases of occupational anaphylaxis follow as a consequence of Hymenoptera stings, insect and animal (mammals or snakes) bites, food, natural rubber latex, chemicals, medications. The causal agent must thus be related to factors and conditions attributable to certain work settings. As a working environment, the agricultural sector is affected mainly and notoriously by these phenomena, due to the risk of the stings and/or bites of arthropods and animals, which can occur during various farming activities. Addressing the issue of anaphylaxis in agriculture, is crucial for this sector, as it is the second greatest source of employment worldwide after service provision, and in which global workforce exceeds more than a third [7]. Agriculture provides a wide range of characteristic subsectors, such as cultivation, breeding, forestry, cereal production, beekeeping, indoor and outdoor activities, seasonality, etc. Therefore, agricultural employees are exposed to multiple conditions, which could be related to an onset of anaphylactic reaction, directly or indirectly. This review aims to look at the anaphylactic phenomenon in a complex economic reality, such as the agricultural sector. It aims to show the main disorders manifested by workers and the principal activities involved. It attempts to highlight the importance of prompt interventions and adequate strategies for management of life-threatening risks on the workplace.

## 2. Materials and Methods

The presentation of this systematic review is in accordance with the PRISMA statement [8].

### 2.1. Literature Research

The research included articles published in the last 10 years on the major online databases (PubMed, the Cochrane Library and Scopus). The search strategy used a combination of controlled vocabulary and free text terms, based on the following keywords: agriculture, farmworkers, allergy, anaphylaxis, anaphylactic shock, exposure, risk. All the various research fields were considered. Additionally, a hand search was undertaken on the reference lists of the selected articles and reviews, so as to carry out a wider analysis. Two independent reviewers read the titles and abstracts of the reports identified by the search strategy. They selected relevant reports according to the inclusion and exclusion criteria. Doubtful papers were discussed with a third researcher whether to be included or excluded. Subsequently, the authors independently screened the corresponding full text articles to decide on final eligibility. Duplicate studies and articles without full texts were eliminated by the authors.

### 2.2. Quality Assessment

Three different reviewers assessed the methodological quality of the selected studies with specific rating tools. We used “International Narrative Systematic Assessment” (INSA) to judge the quality of narrative reviews [9], AMSTAR to evaluate systematic reviews [10] and the Newcastle Ottawa Scale (NOS) to evaluate cross-sectional, cohort studies and case control studies [11]; while the JADAD scale was applied for eventual randomized clinical trials [12].

### 2.3. Eligibility and Inclusion Criteria

The studies included in this review focus on clinical manifestations and pathologies, due to allergic mechanisms, only in exposed farmworkers. Vegetal, animal and chemical allergens can trigger these reactions. No restrictions were applied based on language criteria.

### 2.4. Exclusion Criteria

Works not belonging to the agricultural sector, publications that did not report allergic or immunological disorders, studies about children, adolescents and farmers′ family were eliminated. Similarly, respiratory and dermatologic′ chronic diseases and studies concerning diagnostic methodologies were also eliminated. Editorial articles and individual contributions, as well as purely descriptive studies published in scientific conferences, without any quantitative and qualitative inferences, were also excluded as these were deemed to be of less academic significance.

## 3. Results

The online research yielded 489 references: PubMed (313), Scopus (84) and the Cochrane Library (92). Of these, 398 were excluded because they were deemed not to be related to the allergic disorders in agriculture sector. Of the remaining 91, 55 articles were eliminated as per our exclusion criteria. The remaining, 36 studies were included in this systematic review (Figure 1) Of these, nine were narrative reviews and 27 were original articles. Among the original articles, 12 were cross-sectional studies, while there were also two case series, one case report, four cohort studies, six case-control studies, one pilot study and one experimental study. No systematic reviews were found. Most of the studies were published in the United States, some of which are included in this study (six), most published in 2018 (five). (Table 1).

### 3.1. Reviews

With regards to narrative reviews (Table 2), the INSA score shows an average of 5.1, a median and a modal value of 5, thus indicating an intermediate quality of the studies. The most appropriate methodological review was conducted in Spain (INSA = 6).

We have found nine narrative reviews, which had analyzed allergic disorders in the agriculture sector (9/36; 25%).

Of these, eight studies dealt with respiratory symptoms developed in farmers, in particular asthma and rhinitis (eight/nine; 88.8%); however, the allergens taken into account are the most varied. Storage mites and cow’s dander were the most common causes in this occupational group. Some antigens have been taken into account recently, thus, capsaicin in workers at a red pepper grinding mill [13], tomato (*Lycopersicum esculentum*), cellar spider, turbot (*Scophthalmus maximus*) and olive fruit (*Olea europaea*) [14]. Fungal spores are involved in chronic obstructive diseases and asthma very often [15]. In addition to traditionally identified fungal aeroallergen sources placed in the Ascomycota, some studies have shown a broad diversity of sequences placed in the classes of *Eurotiomycetes*, *Dothideomycetes*, *Sordariomycetes, Saccharomycetes* and *Botrytis cinerea*, a gray mold disease that affects cannabis stems and buds, especially in the cannabis industry and greenhouses [16]. Grass allergens may also become aerosolized in the absence of pollen with mowing, in combination with or without rainfall episodes [17]. Agricultural activity contributes to thunderstorm-triggered asthma by releasing high levels of fungal spores and grass pollen. In widespread pastures north of Melbourne, these activities have clearly been implicated in Australian asthma epidemics. Wheat harvesting is also believed to be involved in other parts of the world where fungal spores are considered the predominant aeroaller [18]. Other antigens potentially involved are biological dust, such as dusts of plant, animal and microbial origin; the innate inflammatory response to biological dust involves several overlapping signaling pathways, including various kinomis (a set of protein kinases) that signal pathways in the lines of the epithelium and monocyte of the human airways and in the monocytic cell lines [19]. In addition, some spices induce the inflammatory reaction based on T-cells after the recognition of the contact of the antigen. In fact, antigen presenting cells (APCs) trigger macrophages, cytokine interleukin 12 (IL-12) and tumor-beta necrosis factor (TNF). Cross-reactivity for protein allergens is another factor that seems to be a significant trigger for the stimulation of these allergic reactions [20]. Finally, Bilò has highlighted that the prevalence of large reactions in beekeepers range from 9 to 31%. Bee venom contains two principal allergens (phospholipase A2, hyaluronidase), and there are some predisposing factors to reactions in beekeepers, such as female gender, having family members with bee venom allergy, more than two years of beekeeping and antihistamine premedication before attending to the hives. The author highlights the 2005 European guidelines, that recommended venom immunotherapy in both children and adults with a history of a systemic, nonlife-threatening reaction (urticaria, erythema, pruritus) and other factors, like occupations or hobbies where the risk of exposure is high [21].

### 3.2. Original Articles

The scores assigned to the original articles have an average value of 5.9, a median of 6 and a modal of 5 (Table 2 and Table 3). This situation amounts to an intermediate quality of the studies; an article from Poland obtained the highest values (New Castle = 8).

#### 3.2.1. Main Diseases

From among 22 studies, five studies (five/22; 22.7%) address the allergic reactions that properly characterize the anaphylactic event due to Hymenoptera stings. These reactions include small localized reactions (redness or swelling around the sting site < 10 cm), large localized reactions (if ≥ 10 cm and persisting up to 24 h) and systemic reactions (skin, gastrointestinal, respiratory and cardiovascular disorders) [22,23]. The face was the body part most frequently stung, and the swelling generally occurred on the eyelids [24]. Factors that predisposed beekeepers to systemic reactions (SRs) included female sex, having a family member with bee venom allergy, more than two years of beekeeping before a SR and premedication with an antihistamine before attending the hives [25]. Many factors may affect the severity of allergic reaction, such as region of residency, type of exposure (natural, occupational or iatrogenic), IgE specific activity (sIgE/T-IgE), etc. Since IgE specific activities are directly related to severity of allergic reactions in venom allergy patients, unlike the low predictive value of Venom-specific IgE (sIgE) in assessment of severity, Guan et al. claim that sIgE/T-IgE may provide more accurate information for diagnosis than serum sIgE [22]. Likewise, Carballo investigated the “tolerogenic role” of honeybee venom (HBV): high serum HBV-sIgG4 levels were associated with less severe sting reactions and lower levels with more severe reactions. Conversely, high concentrations of HBV-sIgE tended to be associated with a greater severity of sting reactions. The quantities of HBV-sIgG4 are also directly related to the exposure indicators (years of beekeeping, number of hives, average of bee stings per year and during the previous year), whereas no relationship with HBV-sIgE, total IgE, or total IgG4 was noted [23]. Mundstedt showed that, after desensitization, all but one patient no longer had an allergic response. In the one remaining case, the severity of the allergy decreased significantly. Interestingly, concern over allergic reactions increased with the number of stings, although they did not cause allergic responses (r = 0.451; *p* = 0.01) [26].

Twelve studies (12/22; 54.5%) address allergic respiratory diseases, such asthma, rhinitis and other respiratory symptoms, related to exposure in the agricultural sector. Of these, three studies focus mainly on asthma. Perotin et al. look at the prevalence of rhinitis (sneezing, rhinorrhea, nasal obstruction, ocular itching), pulmonary (cough, wheezing, dyspnea) and work-related signs and symptoms in crop/vineyard workers [27]; Galli et al. observe the frequency of asthma, rhinitis, respiratory allergy and sensitization profiles related to animal-feeding operations (breeders) and farming environments [28]. In rural women, the prevalence of lower airway symptoms attributable to pesticides was higher, and significantly more prevalent amongst the farm dwellers than town dwellers. In this case, asthma symptom score was associated with low blood cholinesterase (OR 1.93; CI: 1.09–3.44) [29]. The prevalence rate of obstructive pulmonary ventilatory disturbances was higher in individuals with longer exposure (31.7%), whether they were smokers (17.1%) or nonsmokers (14.6%) [30]. In addition, Gascon reports that wheezing was associated with years working in the company (expressed per 10 years), both before (OR 2.55; CI 1.45–4.49) and after (OR 2.99; CI 1.39–6.45) adjusting for other variables [31]. For Patiwael, atopy at baseline was associated with sensitization to occupational allergens, with 65% of the his atopic subjects also being sensitized to the bell pepper plant and 80% of the atopic subjects to at least one of the occupational allergens (i.e., either to the bell pepper plant, to botrytis cinerea or to amblyseius cucumeris), but the duration of employment appeared not to be a risk factor for symptoms [32]. Employees are atopic in the other study conducted by Patiwael of respiratory symptoms triggered by strawberry cultivation (one of the employees exhibited a positive IgE response to dog dander, whereas the other two employees were sensitized to grass) [33].

Six studies dealt with occupational skin diseases (six/22; 27.2%). Śpiewak et al. claim that the incidence of skin disease is four times higher than non-farming occupations and forty times higher than occupational respiratory diseases. In this study, all subjects experiencing skin symptoms complaints of pruritus, succeeded by erythema (54.8%), papules (25.2%), wheals (15.7%), burning sensation of the skin (4.3%), scaling (3.5%), vesicles and edema (each 2.6%). In addition, barely a quarter of cases reported dyspnea, wheezing or chest tightness caused by the exposure to the same aeroallergen of skin symptoms. In almost all cases skin symptoms are located on uncovered skin areas, very few indicated involvement of the entire body. The rate of sensitization was 36.4% in individuals with work-related eczema (EWR) and 57% in work-related urticaria (UWR); in both cases, profiles of sensitization are alike. Interestingly, more than half of UWR (57%) was Type I allergy to occupational allergens, which underlies work-related urticaria, this stands out when considering the rarity of allergic etiology of urticarial [34]. Atopy, together with a family history of any skin disease, along with history of asthma, allergic rhinitis, and eczema (either atopic, allergic or irritant) are relevant risk factors for work-related eczema and urticaria in young farmers. In his study, Oosterhaven suggests glove use as a protecting factor against exposure to allergens, rather than a potential harmful factor due to irritation. The overall occurrence of work-related eczema is 20.5%, of these more than half (64.6%) report eczema in the past 12 months, resulting in a one-year prevalence of hand eczema of 13.2%. Of the workers with eczema who had been patch-tested, 78.6% indicated that they had a contact allergy. The only factor found to be significantly associated with eczema is a lifetime history of atopic dermatitis [35].

#### 3.2.2. Allergens

Nine studies involved allergens of animal origin (nine/22; 40.9%); of these, five studies (five/nine; 55.5%) look at one of the most represented allergens in the agricultural sector, the hymenoptera (honeybee) venom. Other allergens originating from cow, poultry and arthropods were also found. Six studies described vegetal allergens (6/22; 27.2%). Of these, various cereals and pollens are included. Finally, in seven cases (seven/22; 31.8%) generic allergens, such as biological dust, aeroallergens, molds and pesticides were evident.

The most common allergens are pollens (e.g., *Gramineae*; *Compositae*; *Parietaria*; *Betula pendula*; *hazelnut*; *olive*, *cypress*), house dust mites (*Dermatophagoides pteronyssinus* and *Dermatophagoides farinae*), animal dander (dog and cat), feather mix and molds (*Alternaria alternata*, *Aspergillus fumigatus*, *Cladosporium herbarum* and *Penicillium*) [28] and grasses, followed by mixed dusts [36].

The farm work-specific allergens included storage mites Acarus siro, Lepidoglyphus destructor, Tyrophagus putrescentiae, cow epithelium, pig epithelium, horse epithelium and hay, grain and straw dust [34]. Storage mites, specifically Lepidoglyphus, are a typical occupational allergen, for which new sensitization might occur in the first years of a farmers worklife [37] Among swine breeders, high infestation rates of D. farinae and Aeroglyphus robustus in reproductive status and a low to medium presence of Lepidoglyphus destructor, Deuteronymph, Cheyletus eruditus, and Euroglyphus maynei in vital status were found. Among molds and pollens, the mean concentrations of Alternaria, Cladosporium, Fusarium, Epicoccum, Torula, Helminthosporium, Stemphylium, Plantago and Parietaria species were higher outside the farm than inside the pig farms, with intermediate values for inside farms without pigs [28]. Plant dusts were the dominating cause of work-related skin diseases also in hop growers. Plant dusts are complex materials containing variable amounts of allergens, haptens and irritants, as well as live microbes and their products (cell wall components of Pantoea agglomerans stimulate human leukocytes to secrete pro-inflammatory cytokines IFN-γ and TNF-a, which may serve as a "danger signal", initiating sensitization to otherwise harmless environmental allergens and aptens). In some farmers occupationally exposed to thyme dust these developed airborne dermatitis within thirty minutes of work [34]. The study of vineyard workers noted sensitization to pollens gramineae, betulaceae, artemisia, mite D pteronissinus and mold (Alternaria and Botrytis) in more than a third of subjects. Other occupational allergens have been described in vineyard workers or crop farmers: Diphotaxis erucoides in vineyard, spider mite (Tetranychus urticae and Tetranychus macdanieli) in grape farm and citrus red mite (Panonychuys citri) in citrus farms [27]. Recently, the authors investigated new types of allergens, such as triticale [38], fruit, spice, maize and rice pollen [32,33,39], propolis [35] and arthropods, such as Holocnemus pluchei [40].

#### 3.2.3. Agricultural Sectors and Activities 

Almost all articles show an accurate evaluation of farming workplace and activities. Among 22 articles, six publications deal with beekeepers (six/22; 27.2%), five with employees in the production and harvesting of cereals and vegetables, such as triticale, grain, vineyard, sugarcane, maize, rice (five/22; 22.7%), three studies deal with poultry farmers (three/22; 13.6%) and two with greenhouse workers (two/22; 9%).

Indicators of beekeeping exposure, such as years of beekeeping, number of beehives, number of bee stings per year and during the previous year, directly correlate with sensitization to honeybee-venom (HBV-sIgG4) [23]. As is already known, specific farming exposures are known to be associated with a decrease in atopy. This decrease in atopic sensitivity is strongly associated with an exposure to farm animals or animal husbandry during adulthood, with an early life farm exposure and with prenatal exposure. Endotoxin exposure and other microbial agents, including gram-positive bacteria and molds found in high levels in a particular farming environment (cattle, straw and farm milk) appear to be important for providing protection against atopy and atopic asthma. [36] Perotin et al. mark a frequency of occupational symptoms in Champagne vineyard worker, finding a prevalence of work-related (WR) symptoms (11%), of WR rhinitis (6%) and WR pulmonary symptoms (3%), with an increased sensitization to Gramineae, but not to vine pollen. Respiratory symptoms occur mainly in late springs, during the activities of lifting and trellising, and these occur more frequently in women [27]. In Galli et al., the environmental conditions of farms could possibly play a substantial role in influencing the health status of breeders. Exposures related to swine breeding do not increase, and may possibly decrease, the risk of developing allergic diseases (respiratory allergy, asthma, rhinitis, persistent cough). Allergy, asthma, rhinitis, persistent cough and sensitization to grass is lower in swine breeders than in general community living in the same area. In fact, it should be noted that farm sheds have low aeration and are maintained at a constant temperature and humidity to avoid livestock diseases; the poor aeration probably substantially decreases the entrance of pollens in the breeding buildings, and leads to a high concentration of ammonia, produced by swine excreta, which might contrast mold development [28]. Gascon highlights that the percentage of workers reporting wheezing during the last 12 months before the harvesting season was higher among the workers exposed to sugarcane fibers (20.5%), or to other types of dust (16.7%), than in the reference group (11.1%); during the harvesting season, the differences in the FEV1/FVC ratio between exposure groups nearly disappeared [31].

#### 3.2.4. Demographic Characteristics of Involved Workers

Gender was analyzed only in seven studies (seven/22; 31.8%). In most articles, the sample included male workers. Among symptomatic subjects, within a vineyard workers sample, women account for the largest rate of subjects reporting work-related symptoms [27]. Where treated, however, it emerged that women have a lower atopy index than men, although the data reported are not related to age groups and, consequently, to fertile or post-menopausal age. Specific atopic sensitizations are gender-linked as well: men, in fact, are more likely to be atopic to grasses and house dust mites when compared with women [36]. The residential history, studied by some authors, proves to be very interesting, as it emerges that subjects born and raised in highly urbanized contexts, and therefore with probable exposure to a series of predisposing contaminants and irritants, are at greater risk of developing allergic reactions than subjects born and raised in rural environments. The residential history collated by the various studies show that farming populations include native-born subjects, which, in most cases, had been living and working in the same areas. In all the departments examined, subjects had a BMI over 25.0, suggesting general obesity [31]. Smoking and drinking of alcohol was also significantly more common among the farm dwellers. Factors such as age, BMI and household income were not predictors of asthma outcomes, contrary to higher education level, which was generally protective for asthma outcomes [29]. The role of smoking is controversial. The difference in smoking history is reported in two farm populations: compared to breeders (16.7% of smoker and 21.8% of former smoker) [28]. The prevalence of smokers in Perotin et al. is far greater and associated with severity of sneeze, itchy eye, cough and wheezing [27]. Smoking, (chronic) airway symptoms, family history of atopy, on the other hand, were not significantly associated with work-related symptoms [33]. Atopy (OR 4.29), smoking (OR 2.34) and age (OR 0.96) were associated with the development of rhinitis symptoms (*p* < 0.10) [32].

### 3.3. Case Series and Case Report

From three articles, two describe respiratory symptoms in fruit harvesters (peaches) and hops. The third case, on the other hand, reports a particular situation, namely a beekeeper suffering from massive attack of a swarm of bees who is affected by sensory alteration, kidney failure and multiple brain infarcts [41]. In the two studies with farmworkers, the subjects were young, an average age of about 30 years, while in the third case report the beekeeper was older, about 70 years old. The first group exhibited symptoms mainly in the harvest phase and improved with a period of estrangement from work [41,42]. In Reeb, respiratory disease affected not only new workers employed for less than one year (23%) but also in employees with more than 20 years of experience working for their employer (9%) [43]. With regards to the beekeeper, the authors postulate that the tissue damage caused by toxins in honey bee venom (mellitin and phospholipase) induced disseminated intravascular coagulation (DIC), along with systemic anaphylatic reaction-induced hypotension with subsequent ischemia, which resulted in both stroke and renal failure [41].

## 4. Discussion

In the early years of this century the World Allergy Organization (WAO) and the European Academy of Allergy and Clinical Immunology (EAACI), created a new definition of anaphylaxis, which is only based on clinical features, regardless of the underlying physio-pathological mechanism [44]. Clinical criteria have been worked so as to support the anaphylaxis diagnosis [1,2]. Even though similar reactions can be traced back to thousands of years ago [44]. François Magendie, in 1839, was the first to describe the phenomenon, while the Nobel Laureates Portier and Richet, in 1902, decided to adopt the term anaphylaxis, from the Greek words “ana-” (to mean “against”) and “-phylaxis” (“protection”), after observing an immediate severe allergic reaction, and death within 25 min, in dogs injected with a second vaccine dose of a toxin extracted from sea anemones [45,46,47]. The group of immediate symptoms involve skin and mucous membranes (e.g., itching, urticarial/hives, angioedema, morbilliform rash, conjunctivitis, edema of the lips, tongue and uvula), gastrointestinal systems (e.g., nausea, abdominal pain, vomiting, diarrhea), respiratory (e.g., rhinorrhea, nasal obstruction, itchy nose and ear canals, hoarseness, dry cough, stridor, tightness in the throat, dyspnea, wheezing, choking, et cetera), cardiovascular and neurological (e.g., dizziness, weakness, syncope, chest pain, palpitations, diaphoresis, confusion, incontinence, headache, blurred vision, arrhythmias, etc.). In a few minutes, the subject can develop the shock with seizures, non-responsiveness leading to death. Cardiovascular failure can occur without respiratory or other symptoms. Clinical manifestation appears polymorphic and insidious with a range of severity, classified in different scales: the most usual are proposed by Mueller, Ring and Brown [48,49,50]. In about 20% of cases, new symptoms develop, after the acute phase, usually within a mean time of 8–10 h (interval of 0.2–72 h) [4], without a new trigger exposure (biphasic anaphylaxis). More infrequently, persistent anaphylaxis occurs when symptoms persist for 5–32 h [51]. In both these instances, predicting the occurrence is difficult and not always possible; hence, it is recommended to observe the subjects for at least 24 h after the acute phase, and to instruct all those who are prescribed adrenaline as treatment to have two injectors and not one [52].

The pathophysiology commonly underlies an IgE-mediated immune responses as Gell-Coombs Classification type 1 hypersensitivity (immediate hypersensitivity): the antigen-IgE interaction on basophils and mast cells previously sensitized, causes degranulation and release of preformed chemical mediators (e.g., histamine, tryptase, heparin) and de novo syntheses (e.g., leukotrienes, PAF, PGD2), which leads a diffuse contraction of the smooth muscle, resulting in bronchoconstriction, vomiting or diarrhea, for example; as well as an increase in capillary permeability and dilation of blood vessels (e.g., angioedema, urticaria) and a decrease of vascular tone (e.g., hypotension, syncope) [2,4]. At times, biochemical pathways of anaphylaxis are set off by immune cells other than IgE (e.g., IgG, complexes of complement)—traditionally defined “anaphylactoid reaction”—or even through the absence of an immune-mediated cellular activation (e.g., hyperosmolar substances, contrast, anesthetic drugs, physical exercise) [2,3,4].

Despite being mainly based on clinical pattern recognition, in certain situations, the lab analysis may help to confirm the diagnosis of anaphylaxis, through quantifying of the high plasma tryptase levels even after 6 h of its onset, and also by determining the high 24 h urinary methylhistamine concentrations. These tests are more helpful and available in the follow-up care; they are not specific tests for the acute diagnosis and are not performed in case of emergencies [1,4]. As reported by the most recent scientific literature and guidelines, the mainstay treatments of anaphylactic shock are confirmed to be epinephrine as the first line, plus oxygen and fluids. If necessary, antihistamine, steroids and inhaled B-agonist drugs are further treatments [1,53]. Epinephrine use has no absolute contraindications. It can be used even in people with cardiovascular diseases; however, it is often administered late or in sub-optimally doses. The intramuscular adrenaline injection is to be preferred to the intravenous or subcutaneous administration. With regards to the site of injection, the vastus lateralis leg muscle is preferable to the deltoid muscle. The intramuscular dose is 0.2–0.5 mg (1:1000 dilution) and it takes effect within a few minutes; if there are signs of severe reaction, such as throat swelling, airways obstruction and hypotension, persistent further doses have to be administered every 5 min [4]. In community settings, either occupational or non-occupational, it is best to adopt the adrenaline auto-injector (EpiPen[s], Adrenaclick, Emerade, etc.), available in doses of 0.3 mg (0.3 mL) for weight > 30 kg and 0.15 mg (0.3 mL) for 15–30 kg. Susceptible people are furthermore encouraged to wear a medical alert bracelet with the phrase “anaphylaxis, carrier EpiPen (or equivalent)” impressed on it [2]. The increasing frequency recorded globally appears to be conditioned not only by changes in the diagnostic definition and management of the event, but also by many different variables such as triggers, demographic data, geographical areas, seasonal variation, age, gender, atopy and comorbidities (e.g., allergic asthma, cardiovascular disease) [4].

The agriculture sector employs some one billion workers worldwide, or more than a third of the world′s labor force, and accounts for approximately 70 per cent of child labor worldwide. It is also the largest sector for female employment in many countries, especially in Africa and Asia [7]. Agriculture represents one of the main sectors at risk of anaphylactic reaction, following the exposure of farmers to several elicitors specific to the farming environment. Stings and bites by invertebrates—such as *Hexapoda* (Hymenoptera) *Arachnida* (e.g., spiders, scorpions), *Acarina* (ticks, mites), *Chilopoda* (centipedes) etc.—and vertebrate animals (e.g., snakes, horses)—are the main risks for the exposed population. For example, farmers who spend time in fields, forested areas, tropical jungles and caves may be bitten by snakes or pine processionary caterpillars; albeit, insect stings, together with spider, tick and scorpion bites, are common experiences of agricultural workers who are involved in crop husbandry (e.g., goatherds, shepherds, livestock farmers), the harvesting of fruit, tubers, grain or the storage and handling of such agricultural products [5,7].

In the recent EAAI consensus statement on the management and prevention of occupational anaphylaxis is has been reported that beekeepers, gardeners, farmers, and forestry workers are the professionals most frequently affected by occupational venom allergy [5].

The type of insect, as well as risk factors related to the host and exposure may play a role in developing an allergic reaction or for the severity of anaphylaxis in farm workers. Anaphylactic reactions were reported in 40% of 30 workers with occupational allergic urticaria caused by *Thaumetopoea pityocampa* (pine processionary caterpillar) [54]; thus, pinecone or resin collectors, woodcutters, farmers and stockbreeders were the most frequently and severely affected workers [7]. Greenhouse workers may develop a bumblebee venom allergy, because, in greenhouses, bumblebees are used for the pollination of plants [55,56]. Hymenoptera stings are specifically recognized as a high occupational risk for beekeepers (14–43%) [21,57]; the intense exposure to stings cause honeybee venom (HBV) to be the dominant allergen [22,23]. Repeated (one or more) previous stings were found to increase the risk for subsequent severe anaphylactic sting reactions [58], and also of anaphylactic shock in workers with a previous episode of anaphylaxis [22]. Some authors point bunch of patients with a history of large localized SR develop systemic symptoms when re-stung [23,59] and they recommend paying attention to large localized SR with close follow-up and providing them with the emergency kit. Guan et al. [22] whilst observing an iatrogenic exposure (6% by honeybee apitherapy), and a natural exposure to honeybees during travel to rural surroundings for work or tourism (nearly of half HBV allergy by hiking) underline the importance of recognizing a previous allergic SR, making differential diagnosis with “asymptomatic sensitizations” (having positive venom sIgE, and usually high total IgE, without honeybee sting history)—caused by cross-reacting carbohydrate determinants inside many plant allergens [60,61,62].

On the other hand, although beekeepers are more likely to be stung and, thus, experience more serious reactions, it has been established that multiple stings may act as natural immunotherapy to HBV, so that only a limited percentage of beekeepers develop severe anaphylaxis [23]. The tolerogenic role of HBV-sIgG4 presented in Carballo et al. corroborate those of previous studies [63,64,65,66]. A careful review of the clinical history of anaphylaxis plays a significative role, making it a critical element to confirm diagnosis of the allergic reaction to HBV, along with positive venom extract skin test and/or serum specific IgE (sIgE) [22]. It has been observed that performing HBV serum specific IgE quantitative measurements is positive in about 40% of adults with recent honeybee stings [67]. This has a low predictive value in the assessment of the severity of the reaction [23]. In an effort to ameliorate predictiveness by providing accurate information for the diagnosis, several studies show that allergy specific activity (the allergen-specific IgE to total IgE ratio; sIgE/T-IgE analysis) directly correlates to the severity of the allergic reactions in venom allergy patients [22,68]. In a group of (ex) beekeepers, the onset of allergic or anaphylactic SR is the main reason for quitting beekeeping activities [35]. Long-term care of farmers with a Hymenoptera venom allergy includes patient education (allergen avoidance, course of action on re-sting) and an emergency kit prescription that should be mandatory (epinephrine by intramuscular auto-injection; EpiPen). Moreover, the opportunity could be considered to administer specific treatment as venom immunotherapy [22]. According to the literature results, a maintenance dose of 100 μg can prevent a systemic reaction in 75–95% of re-stung patients, suggesting a better effect with higher maintenance dose [69]. 

Farmers are exceedingly exposed to various agents—mostly through inhalation and dermal contact, more rarely through ingestion—which act as elicitors, inducing hypersensitivity reactions on the respiratory and/or skin system. Dermatological problems are four times more common in farmers than in non-farming workers, and occur forty times more than occupational respiratory diseases [34,70]. Rhinitis, sinusitis, asthma, asthma-like syndrome, hypersensitivity pneumonitis, chronic bronchitis and chronic obstructive pulmonary disease have been related to animal-feeding operations and farming environments [28].

It is not always easy and immediate to claim that illness and symptoms related to occupational exposures are definitely due to an underlying anaphylactic hypersensitivity mechanism. It often presents as similar illness due to irritative/inflammatory responses of the respiratory tract (asthma, rhinitis, cough, etc.) or even more often of the cutaneous (eczema, contact dermatitis) system. Additionally, the prevalence and severity of occupational respiratory symptoms are associated with particular sensitization profiles, occupational activities, socio-demographic characterizes, weather conditions, environmental and exposure features [27,28,71]. According to Perotin et al., the severity of work-related pulmonary symptoms does not relate to respiratory functional tests results. 

Rhinitis was defined as the occurrence of two or more nasal symptoms or one nasal symptom and one eye symptom. Work-related symptoms were defined as being present during working hours with improvement of the symptoms during the evenings, weekends and/or on holidays [27]. The clinical diagnosis of asthma is confirmed by assessing the history of respiratory symptoms and presence of expiratory airflow limitation on spirometry test. Respiratory allergy refers to the presence of respiratory allergic disease (rhinitis and asthma) and a positive skin prick test (SPT) reaction or presence of specific IgE [28]. Sensitization or atopy is performed by skin prick test (SPT), showing one or more positive reactions (wheal diameter ≥ 3 mm) to tested allergens and positive (histamine) and negative (saline) controls [27,28,32,34,37]. Respiratory function is tested for each subject with respiratory symptoms (FEV1/FEV6; FEV1/FVC. The nasal cytology pattern is performed to assess changes in nasal epithelium exposed to allergens, irritants or inflammatory agents of farmers and to define the allergic pattern [28].

The selected studies of work-related skin diseases refer mainly to term “eczema” or “urticaria”, since a detailed insight into the underlying mechanisms is not always possible—e.g., differentiating irritant or allergic etiology in airborne dermatitis. In fact, when analyzing the history of contact dermatitis (CD) differentiating between irritative contact dermatitis (ICD) and allergic contact dermatitis (ACD) would be hardly possible based merely on the patient’s history, even for a specialist [34]. In farmers with UWR has been observed a significantly higher frequency of plant dust allergy and a relatively high coincidence of grain dust-related skin disease with respiratory symptoms; these findings may give an indication of a systemic allergy, the same as in contact urticaria syndrome; single cases of immunological contact urticaria caused by cereal allergens have been reported previously [34,72]. In farmers with HE and positive patch-test, contact allergy occurs (78.6%); atopic dermatitis, indeed, is a well-known risk factor for HE [35].

When focusing on the risk factors, it is important to take a precise assessment of the working conditions. Working in a wet work occupation or glove-wearing during the activities can contribute to the development of hand eczema; on the other hand, the skin of farmers who do not (always) wear gloves might be regularly exposed to contact allergens.

Agricultural workers are exceedingly exposed to various allergens, most of which are commonly present in various agricultural sectors, while others are strictly linked to specific activities within the farming environment. The nature of the farming environment with both livestock and crop production makes it a favorable environment for both increased and decreased susceptibility to allergy in farm dwellers [36]. Storage mites (SM), specifically *Lep d*, is an allergen typically belonging to the agricultural setting. Although SMs are found outside farming the concentration levels in farms, especially in barns and stables (infestations in grain, straw, and hay storages) are substantially higher compared to urban households. Sensitization and allergies have been found primarily in occupationally exposed individuals such as farmers, millers, grain and meat production workers, where storage mite exposure is high. Moreover, in settings such as barns and stables, characterized by a moderate-high endotoxin exposure risk, alongside an elevated *Lepidoglyphus destr* (*Lep d*) exposure, are associated with increased new-onset sensitization to Lep d, and significantly less loss of *Lep d*-IgE sensitization. General SM sensitization-levels may be affected by house dust mite (HDM) exposures and sensitization, due to cross reactivity between HDM and storage mites. While for the common allergens, exposure levels presumably remain as relatively constant in adulthood as in childhood and adolescence, it is probable that the levels of general storage mite exposure increase considerably for young adults who start working as farmers [37]. In a study by Śpiewak et al., the leading causes of urticaria (UWR) and eczema (EWR) work-related among vocational farm students are grain, hay and straw dust, storage mites (*A. siro*, *L. destructor*, *T. putrescentiae*) and house dust mites *D. pteronyssinus* sensitization. The results corroborate the widely held knowledge in the literature recognizing plant dusts as the principal cause of WR skin symptoms and diseases among farm workers [34,73,74,75]. The assessment of working features, on-site inspections, occupational history, injuries and near-miss logbooks crucial to better address preventive strategies and measures. The prevalence of rhinitis in crop farms is described in 13% to 41% of farmers, symptoms of asthma in 3% to 7%, depending on the studies performed in farmers from different European countries and exposed to different cultures (fruits/vegetables, flowers, grain) [76]. Previous European studies described WR symptoms in 25% of fruit farmers [77]. 

A study comparing grape farmers to a control population in Greece showed that grape farming was a risk factor for occupational sneezing (OR 2.9), rhinorrhea (OR 2.5), cough (OR 3.7) and dyspnea (OR 3.8) [77]. Spiewak described occupational sneezing and rhinorrhea in 16% of a population of crop farmers, exposed to grain dust. Cough and dyspnea in 9%. The environmental conditions of the farms could play a substantial role in influencing the health status of breeders [73].

Growing up in a farm environment is a well-established protective factor against atopic sensitization to common allergens in adulthood, while the prevalence of atopy is not associated with current farm living. Working as a farmer in young adulthood may also provide protection against incident sensitization and the persistence of existing sensitization, especially to pollen allergens [36,37]. This protective effect of exposure might be related to the “hygiene hypothesis,” which maintains a lower prevalence of allergic diseases in people who reside in the countryside. In order to explore the specific farming exposures associated with a decrease in atopy, it is thought that the major stimulation of the Th1 immune response in young farm children, due to an increased exposure to endotoxin on farms with livestock production, might result in a down-regulation of Th2 helper cell immunity necessary for allergic responses in children and adults to environmental antigens (hygiene hypothesis) [36].

The low prevalence of asthma, rhinitis, and respiratory allergy in native-born SB in the present study is in agreement with other studies carried out in European swine farmers. Accordingly, the prevalence of sensitization to common allergens in native-born breeders (35.9%) is similar to that reported in agricultural workers in other European studies (13% to 64%) [76,78,79]. To explain these findings, it should be noted that farm sheds have low aeration, and are maintained at a constant temperature and humidity, to avoid livestock diseases. The poor aeration probably substantially decreases the entrance of pollens in the breeding buildings, and leads to a high concentration of ammonia, produced by swine excreta, which might contrast mold development. These findings are in apparent contrast with previous studies showing that SB increased the risk of respiratory diseases and symptoms and accelerated decreased lung function with aging, possibly owing to differences in sampling time and places, measurement technique, equipment and diagnostic criteria. The environmental conditions of farms could play a substantial role in influencing the health status of breeders [28].

Specific allergens appear to be more important for an increase in or protection from allergy that is related to the timing of farming exposures [36]. Residential history changes from childhood to adulthood, which also appear to vary by gender, may result in triggering the sensitization onset in those individuals who lived in a polluted urban environment, unlike those who have always lived in a rural setting [36]. The results are consistent with other studies. Ernest et al. observe that a decrease in sensitization found in the farming group was significantly different only for women (*p* < 0.001) [80]. Furthermore, Hoppin et al. noted that farming women in the Agriculture Health Study who grew up on a farm, and who did not apply pesticides, had the lowest risk of atopic asthma compared with women who never grew up on a farm. For men, there was a trend for an increased risk of atopy to HDM for those who currently lived on a farm [81].

Recent studies have shown that working on a farm seems to have a protective effect against ongoing or newly developing sensitization to common allergens, independent of childhood farm exposure [82,83]. New onset, as well as persistence of sensitization with farm work, can be explained by the exposure to farm dust during adult farm work, in contrast to the common allergens exposure levels that presumably remain as relatively constant in adulthood as in childhood and adolescence. In case of hyper susceptibility conditions, which could occur in case of comorbidities, such as respiratory and cardiovascular pathologies, it appears that it is very important to avoid any further exposure to prevent a potentially life-threatening response [2].

No study has dealt with the possible impact of physical exercise and fatigue as possible risk factors in triggering anaphylaxis. Farmers are often engaged in tasks that require significant physical efforts. Exercise-induced anaphylaxis attacks are not consistently elicited by the same type and intensity of physical activity in each patient. Co-factors such as foods, alcohol, temperature, medications (i.e., aspirin and other no steroidal anti-inflammatory drugs), humidity, seasonal changes and hormonal changes are important in the precipitation of attacks. Hence, work-related exposures in the agriculture sector are extremely complex and difficult to measure. All the studies examined have some limitations, such as low response rate, resulting in non-response bias, no temporal relation between the beginning of allergic disease and the start of employment, the “healthy worker” selection bias, because some workers who had developed respiratory symptoms at the beginning of their employment might have withdrawn from this work, which could result in a low prevalence of allergic diseases and a relatively small sample that might have underpowered the study.

## 5. Conclusions

Agriculture is widely acknowledged as being one of the most hazardous of all employment sectors throughout the world, causing many occupational accidents and diseases in farm workers every year. Elicitors and triggering factors attributable to the agricultural environment and activity may lead to anaphylactic reactions. This study purposes to improve awareness of such a risk within the articulated sector with wide-ranging profiles both in terms of employment and of enterprise. Within the multiple agriculture sub-sectors, including the variability of production processes (ranging from the highly mechanized and large scale operations (intensive agriculture), to the less mechanized and more labor-intensive farming (extensive agriculture), and to the biological ones, their seasonal trend and the complexity of multiple tasks involved, all determine different exposure conditions that are associated to precise risk profiles. Furthermore, awareness of anaphylaxis risk in particular groups of workers, such as women and migrant workers, should be of particular concern. Hence, the need to assess the potential risk of allergic anaphylaxis linked to individual activities, so as to be able to suggest accurate prevention and protection measures, as well as management strategies, specifically addressed to all specialists who deal with occupational health and safety.

## Figures and Tables

**Figure 1 ijerph-17-04921-f001:**
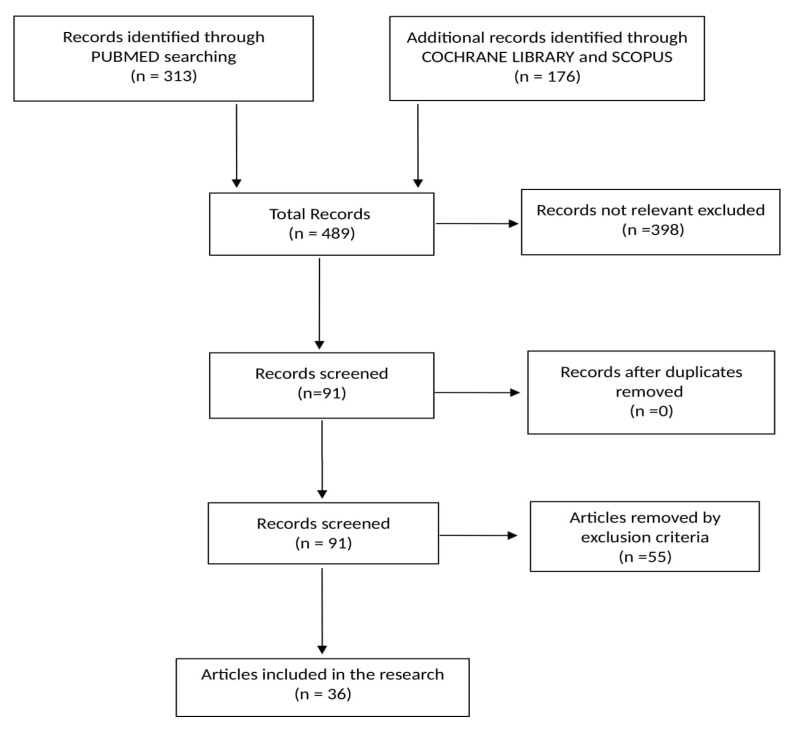
Flow chart of the bibliographic research.

**Table 1 ijerph-17-04921-t001:** The studies included in this systematic review, (in alphabetical order).

First Author	Year	Study	Key Messages
Al-Maneea	2013	cross-sectional	Exposure to animal products (dust, pigeon, dog, cat, turkey) exacerbates allergic asthma
Ballal	2016	narrative review	Storage mites and cow’s dander are the commonest causes of occupational rhinitis in this occupational group
Bilò	2012	narrative review	Allergic beekeepers represent unique populations for epidemiological, venom allergy immunopathogenesis and mechanism studies
Bobolea	2011	case control	Spiders represent an important but as yet unexplored source of indoor allergens, especially in agricultural environments
Carballo	2017	case-control	High concentrations of HBV-sIgG4 associated with less severe sting reactions in beekeepers. HBV-sIgE tended to be higher in beekeepers with more severe reactions, while total serum IgE was lower in beekeepers than in controls.
Çelıksoy	2014	cross-sectional	The risk of a severe reaction to bee stings increases with the degree of exposure and beekeepers are at the highest risk
Elholm	2018	cohort Studies	Storage mites can be typical occupational allergen for which new sensitization might occur in the first years of a farmer’s work life.
Ercilla-Montserrat	2017	experimental study	It is possible to recirculate the air of the i-RTG to the building, thus to converting the system into an Bi-RTG without posing health risks due to allergies for the building users if the biological air quality is monitored and the corresponding preventive measures are taken, even with the installation of air filters.
Galli	2015	case-control	The prevalence of allergic diseases is lower in people who reside in the countryside and therefore are exposed to bacterial, parasitic, and viral infections during childhood
Gascon	2012	cohort study	During the harvesting season, the prevalence of wheeze and eye problems almost doubled in workers exposed to bagasse and other types of dust
Green	2018	narrative Review	Health and safety precautions used in the landscape industry can be used by gardeners, landscapers and arborists to reduce or eliminate exposure to biological hazards
Green	2018	narrative Review	Occupational fungal exposures include a much broader diversity of fungi than once thought
Green	2018	pilot study	Potential exposure to microbiological hazards, such as Actinobacteria and cannabis fungal, in workers that harvest, bud strip or hand trim organically produced cannabis.
Guan	2016	case-control	The use of sIgE/T-IgE results is a useful diagnostic parameter in determining honeybee venom allergy
Harun	2019	narrative review	Wheat harvesting has implicated in respiratory symptoms, with fungal spores as the predominant aeroallergen
Jain	2012	case report	Venom bee stings can cause neurological complications (stroke, seizure, aphasia, dysarthria, apraxia, ataxia and coma)
Merget	2016	case control	Triticale allergy may occur as a distinct allergy in farmers
Munstedt	2010	cross-sectional	Desensitization can result in a complete absence of symptoms after re-exposure to bee stings
Ndlovu	2014	cross-sectional	Pesticide exposure among women farm workers is associated with increased risk of ocular nasal symptoms and an elevated asthma symptom
Nordgren	2018	narrative Review	Working and/or living near CAFOs (concentrated animal feeding operations) is a risk factor for development of various respiratory diseases due to various contaminants, like biological dust, pesticides and zoonotic pathogens
Nordgren	2016	narrative Review	In agricultural industry, occupational exposures to bioaerosols and inorganic aerosols lead to increased risk for lung disease amongst workers
Oosterhaven	2019	cross-sectional	The prevalence of hand eczema in beekeepers was higher than in the general population, but a small subset of beekeepers suffered it
Patiwael	2010	cohort study	Pollen from bell pepper plants cultivated in greenhouses cause occupational allergic disease
Patiwael	2010	cross-sectional	Allergic symptoms attributable to the workplace are present among a proportion of strawberry greenhouse employees
Perez-Calderon	2017	case-series	Sensitization to no-pollen tree structures, such as peach leaves, may cause occupational respiratory symptoms.
Perotin	2015	cross-sectional	Work-related respiratory symptoms are frequent and associated with a sensitization to gramineae and with activities performed close to vine in late spring
Pesonen	2020	cross-sectional	Main causes of CU and PCD in Finnish workers are animal dander, grain, NRL, and plant- and animal derived foods.
Quirce	2011	narrative review	Asthma rapidly accelerated with the advent of new technologies introducing a spectrum of new agents into the workplace
Ramavovololona	2014	cohort study	Major allergens β-expansins, profilin and polygalacturonase were characterized both in maize and rice pollen
Reeb-Whitaker	2014	case series	Occupational exposure to hop dust is associated with respiratory disease, especially in hop workers
Rennie	2015	cross-sectional	There are associations between atopic sensitization and farm living that appear to vary by sex. Specific allergens appear to be more important for an increase in or protection from allergy that is related to the timing of farming exposures.
Richter	2011	cross-sectional	Factors that predisposed beekeepers to allergic reactions are female gender, having a family member with bee venom allergy, more than 2 years of beekeeping and premedication with an antihistamine
Sabino	2012	case control	This study confirmed the presence and distribution of Aspergillus in Portuguese poultry and swine farms
Spiewak	2017	cross-sectional	Work with hops is the kind of plant production associated with most frequent skin diseases, followed by work with grain, hay and straw. Skin prick tests do not correlate well with referred symptoms.
Upadhyay	2019	narrative review	People involved in spice agriculture and food industries are at greater risk to long and short-term respiratory issues
Viegas	2013	cross-sectional	Poultry farm workers are more prone to suffer from respiratory ailments and this may be attributed to higher concentrations of particulate in the dust

**Table 2 ijerph-17-04921-t002:** Reviews and cohort studies, case-control, case series, case report, pilot study, experimental study, with their relative scores.

First Author	Included Subjects	Allergic Disease	Allergens	Category Workers	Scores
Ballal SG		Rhinitis	various	Farmers, florists, greenhouse, animal, grain handlers	I.5
Bilò MB		Anaphylaxis	Bee Venom	Beekeepers	I.5
Green BJ		Respiratory disease	Fungal bioaerosols	Various	I.5
Green BJ		Respiratory, dermatologic	Pollen, Arthropods, Bioaerosol	Gardeners, Horticulture, Greenhouse	I.5
Harun DS		Asthma	Pollen	agricultural activities	I.5
Nordgren M		Respiratory disease	Biological Dust, Bioaerosols	Various	I.5
Nordgren M		Respiratory disease	Aeroallergens	Farmers	I.5
Quirce S		Asthma	Various	Various	I.6
Upadhyay E		Respiratory, dermatologic	Arthropods	Spice agricultural workers	I.5
Bobolea I	1 case, 5 controls	Asthma	Arthropods	Cereal workers	N.6
Carballo I	158 cases, 465 controls	Anaphylaxis	Honeybee Venom	Beekeepers	N.6
Elholm G	not specified	Sensitization	Arthropods	Farmer, Millers, Cattlemen	N.7
Ercilla-Montserrat M	not specified	Respiratory disease	Biological dust	Greenhouses worker	n.a.
Galli L	101 cases, 82 controls	Respiratory disease	Aeroallergens	Swine breeders	N.6
Gascon M	N.74	Respiratory, ocular disease	Biological dust	Sugarcane workers	N.6
Green BJ	not specified	Respiratory, dermatologic	Microbiological hazard	Farm Cannabis workers	n.a.
Guan K	54 cases	Anaphylaxis	Honeybee Venom	Beekeepers	N.3
Jain J	N.1	Stroke	Bee Venom	Beekeepers	n.a.
Merget R	1 case, 4 controls	Asthma	Triticale	Farmers	N.6
Patiwael JA	N.322	Respiratory disease	Pepper Pollen	Greenhouses	N.7
Perez-Calderon	N.37	Respiratory disease	Peach tree	Peach Crop	n.a.
Ramavovololona	N.65	Respiratory, dermatologic	Pollen	Maize, Rice Crop	N.5
Reeb-Whitaker	N.57	Respiratory disease	Lupulus	Hop workers	n.a.
Sabino R	47 cases, 28 controls	Respiratory, dermatologic	Aspergillus	Swine, Poultry workers	N.7

**Table 3 ijerph-17-04921-t003:** Cross articles that were included, together with their relative scores, in alphabetical order.

First Author	Included Subjects	Allergic Disease	Allergen	Category Workers	Score
Al-Maneea	N.10	Respiratory disease	Animals	Not specified	N.5
Çelıksoy MH	N.301	Anaphylaxis	Bee Venom	Beekeepers	N.5
Munstedt K	N.63	Anaphylaxis	Bee Venom	Beekeepers	N.5
Ndlovu V	N.211	Asthma	Pesticide	Not specified	N.6
Oostarhaven	N.833	Dermatitis	Various (propolis)	Beekeepers	N.7
Patiwael JA	N.75	Respiratory, dermatologic	Strawberry Pollen	Greenhouses	N.6
Perotin JM	N.307	Respiratory disease	Aeroallergens	Vineyard workers	N.5
Pesonen M	N.570	Dermatitis	Cow, grain, flour dust	farmers, livestock workers, gardeners	N.6
Rennie DC	n.11982	Atopic/allergic reaction	Aeroallergens	Farmers	N.7
Richter AG	N.852	Anaphylaxis	Bee Venom	Beekeepers	N.6
Spiewak	N.440	Dermatitis	Biological dust	students of agriculture	N.8
Viegas S	N.46	Respiratory disease	Poultry dust	Poultry farmworkers	N.5

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
