# Peer review of "Allergic Anaphylactic Risk in Farming Activities: A Systematic Review"

_ijerph, 2020, doi:10.3390/ijerph17144921_

Round 1

Reviewer 1 Report

Dear Author, this is very interesting read. There are quite a number of grammatical errors and significant work needs to go into the standardization of the refences. I think that a summary table or bullet points of your findings would be beneficial. The tables are illegible and need to be re-done. Please see my other comments below.

Line 21-22 needs to be restructured.

Grammatical error 23-24, remove “have been”

Please consider, don’t include the number of reviews looked at, instead highlight some of your findings. It would be much more engaging for the reader.

What do you mean by “demographic dates”, should this be ages?

Line 56-57 requires attention, is confusing.

Please address line 68, exercise? Fiver?

Consider putting Table 1 into supplementary material.

Table 2 and 3 are illegible

What is “organic dust”?

In Table 3, what do you mean by “Cross articles”? More information in the table legend would be useful.

Line 193, “v” before Likewise.

Can you explain, or hypothesize why premedication with anti-histamine is a risk factor?

Full words for HBV not mentioned until line 425, should be mentioned earlier.

Et al. should be in italics.

Line 272 full stop missing.

Line 280, consider revising the word “sight” here.

Line 281, pubblications spelled incorrectly.

Re-word line 516. “widely knowledge” does not make sense.

Line 534 – full stop missing.

References section.

Significant word needs to go into the standardization of the references.

Missing full stops at end of references.

Reference 5 is incomplete.

A table should be included to summarize your most important findings.

Author Response

Dear reviewer,

your required changes are underline in the text in Yellow. We have correct the references, the English Language and new tables (in particular Tab.1 con key messages).

Kind regards

Reviewer 2 Report

The manuscript needs considerable language editing.

The introduction is dispersed and does not provide the reader with a focused background. Agriculture, which is the main focus of the work only appears sparsely, later in the introduction section. This needs to be restructured.

All the figures are of poor quality

Line 23: authors analyzed 36 articles….

Line 40: What changes are seen in these organs (systems)?

Line 41-45: “Because of its heterogeneous…… is still uncertain [4].” This is extremely long and complex. I suggest the authors split into shorter sentences to make the meaning clear.

Line 45-46: “Whereas in public health terms, it is not considered a common cause of death [1], a life-threatening anaphylactic risk is ubiquitous”. It is not clear what the authors are saying here. Please rephrase.

Line 48: populations, especially regarding the…

Line 50: it ranges from 0.05 - 2%

Line 75-76: Agriculture provides a wide range of several characteristic subsectors

Line 94: “they”…who?

Line 280: “Almost all articles sight” what does this mean?

Author Response

Dear reviewer,

your required corrections are underline in the text in Green. We have correct introduction, the references, the English languages and we have improved the tables.

Kind regards

Round 2

Reviewer 1 Report

Please revise first sentence, "Allergic disorders in the agricultural sector are some of the most hazardous risks..."

By 'organic dusts', do you mean 'biological dusts'? 

Line 182, United States, there should be an 's' at the end of States

Line 182, consider re-phrasing

Line 234 should be 'without'

Again, what does 'sperimental', this is found in line 181 and in table 1. 

You have written 'molds' and 'moulds', should be 'moulds'. 

All latin names should be in italics, as well as et al. 

Line 421, there is a capital letter in the middle of the sentence.

The manuscript, should be re-read including all the track changes, there are still quite a number of grammatical errors. 

Author Response

Dear Reviewer,

we made the required corrections.

Kind regards

Reviewer 2 Report

Minor language edits may still be required.

Author Response

Dear reviewer,

we made the required corrections.

Kind regards
